# Blood–Brain Barrier Transport of Transferrin Receptor-Targeted Nanoparticles

**DOI:** 10.3390/pharmaceutics14102237

**Published:** 2022-10-19

**Authors:** Maj Schneider Thomsen, Kasper Bendix Johnsen, Krzysztof Kucharz, Martin Lauritzen, Torben Moos

**Affiliations:** 1Neurobiology Research and Drug Delivery, Department of Health Science and Technology, Aalborg University, 9220 Aalborg, Denmark; 2Department of Health Technology, Technical University of Denmark, 2800 Kgs. Lyngby, Denmark; 3Division for Science & Ethics, Danish National Center for Ethics, 2750 Ballerup, Denmark; 4Department of Neuroscience, Faculty of Health Sciences, University of Copenhagen, 2100 Copenhagen, Denmark

**Keywords:** antibody, blood–brain barrier, endosomal, liposome, nanoparticle, targeting, transferrin

## Abstract

The blood–brain barrier (BBB), built by brain endothelial cells (BECs), is impermeable to biologics. Liposomes and other nanoparticles are good candidates for the delivery of biologics across the BECs, as they can encapsulate numerous molecules of interest in an omnipotent manner. The liposomes need attachment of a targeting molecule, as BECs unfortunately are virtually incapable of uptake of non-targeted liposomes from the circulation. Experiments of independent research groups have qualified antibodies targeting the transferrin receptor as superior for targeted delivery of nanoparticles to BECs. Functionalization of nanoparticles via conjugation with anti-transferrin receptor antibodies leads to nanoparticle uptake by endothelial cells of both brain capillaries and post-capillary venules. Reducing the density of transferrin receptor-targeted antibodies conjugated to liposomes limits uptake in BECs. Opposing the transport of nanoparticles conjugated to high-affine anti-transferrin receptor antibodies, lowering the affinity of the targeting antibodies or implementing monovalent antibodies increase uptake by BECs and allows for further transport across the BBB. The novel demonstration of transport of targeted liposomes in post-capillary venules from blood to the brain is interesting and clearly warrants further mechanistic pursuit. The recent evidence for passing targeted nanoparticles through the BBB shows great promise for future drug delivery of biologics to the brain.

## 1. Introduction

The brain harbors a vascular barrier system consisting of the blood–brain barrier (BBB) and the blood–cerebrospinal fluid (CSF) barrier. Together, they limit the passage from the bloodstream into the brain parenchyma of virtually any drug unless highly lipophilic or smaller than approximately 70 Daltons [1,2,3,4]. This is preferable from a physiological perspective, as it allows to sustain integrity and maintain stable extracellular concentrations of solutes in the brain with minimal influence from fluctuations in blood. It is also advantageous from a toxicologic perspective, as the vascular barriers form a strong defense that prevents the entry of unwanted exogenous substances and pathogens [1,2,3,4]. 

However, the presence of the brain barrier system is inconvenient from a pharmacological perspective as many of the existing and forthcoming drug candidates, e.g., polypeptides, or genetic material-based pharmaceutics like siRNA or cDNA, are unable to pass the brain barriers [5]. This is unfortunate as research of recent years has identified several targets putatively amendable for the treatment of CNS diseases, providing that such biologic therapeutics (“biologics”) can enter the brain [6,7]. Consequently, current therapeutics attempting to treat neurodegenerative disorders like Alzheimer’s and Parkinson’s disease, and psychiatric diseases, e.g., schizophrenia and endogenous depression, where the vascular barriers are supposedly intact [8], are pharmacologically restricted. 

The discovery of the selective expression of targetable proteins by brain endothelial cells (BECs) has changed the scene and provides new optimism, as several biologics conjugated to targeting antibodies are now amendable for uptake and transport across the BECs [2,5,6]. The use of targeted antibody-conjugated nanoparticles for drug delivery to the brain denotes an interesting alternative to the use of biologics conjugated directly to antibodies [9,10]. A major advantage of nanoparticles is their omnipotence to encapsulate potentially any drug of interest with minimal restraints on their size [9,10]. From the first studies on BBB transport performed more than two decades ago [11,12,13], the exploration of targeted liposomes and other types of nanoparticles, e.g., polymeric nanoparticles, gold nanoparticles, and magnetic nanoparticles, has progressively continued [9]. A recent peak in this discovery was the identification of transport of transferrin receptor-targeted liposomes into the brain, recently reported using real-time two-photon microscopy in vivo. Interestingly, transferrin receptor-targeted liposomes were found to mainly undergo transport into the brain via an unexpected route, i.e., across endothelial cells of post-capillary venules, and not via brain capillaries [14].

Here, we review the most successful attempts made to enable targeted uptake and transport of nanoparticles across the BBB. As most studies target the transferrin receptor, our delineation of the existing literature mainly addresses this receptor. We describe how specific proteins expressed by BECs enhance the binding and uptake of antibodies from the circulation. Next, we cover how targeted nanoparticles can undergo specific binding and uptake, when conjugated with antibodies weakened in affinity or avidity. We also discuss the therapeutic use of targeted nanoparticles in conditions with brain pathology.

## 2. Passaging of Large Molecules through the Blood–Brain and Blood–CSF Barriers

The BBB proper consists of endothelial cells connected by tight junctions. The blood–CSF barrier is formed by choroid plexus epithelial cells also connected by tight junctions, but opposed to the BBB; the capillaries of the blood–CSF barrier are leaky, meaning that solutes of the plasma diffuse into the extracellular space of the choroid plexus where the epithelial cells selectively transport molecules to the CSF [2,3,14]. From a quantitative drug delivery perspective, passaging across the BBB is by far the most important as the brain microvasculature has a surface area thousand-fold higher than that of the choroid plexus. This allows drugs to enter the entire brain while transport across the blood–CSF barrier is restricted to the ventricular system [2,3,15].

The BBB prevents large molecules and particles in blood plasma from entering the brain (Figure 1). This includes entry via the paracellular space between the endothelial cells where tight junctions limit diffusion from blood to brain [1,2,9]. To enable nutrient uptake while preventing the influx of unwanted substances, the BECs express nutrient transporters for, e.g., amino and fatty acids, monosaccharides, vitamins, and essential ions and metals [1,2,3,4]. In contrast, the transport of large molecules of the plasma, like albumin and IgG, is diminutive, e.g., intravenous injection of non-immune IgG in the adult rat is limited to as little as 0.03% of the injected dose, which can be surpassed more than ten-fold by injecting anti-rat transferrin receptor-targeted IgG (OX26) [16]. This can also be observed at the ultrastructural level, where a limited number of transporting vesicles occurs in the cytosol of the BECs compared to endothelial cells of other non-fenestrated capillaries, e.g., those of skeletal muscles. For the same reason, the chances of obtaining transport through the brain endothelium of large constituents like nanoparticles are also predictably low, unless nanoparticles are made targetable to nutrient transporters (see next paragraph).

## 3. The Transferrin Receptor as Target for Drug Delivery

The first indicator of specific uptake and transport of a plasma protein came from observations showing that the brain has a high binding capacity for transferrin, the transporter of the essential metal iron [17]. Later, the identification of a specific binding protein for transferrin, the transferrin receptor (a.k.a. transferrin receptor 1), was identified on BECs [18]. Except for a few other organs, e.g., gonadal cells, the expression of the transferrin receptor by BECs is different from capillaries of organs elsewhere in the body, which do not express this receptor [18,19,20,21,22,23].

Among the large proteins present in blood plasma, transferrin stands out because of its potential for binding to the transferrin receptor of the BECs [9,24]. The quantitative uptake of iron-containing transferrin by the receptor was first addressed thirty years ago in seminal studies by Morgan and co-workers, who co-examined brain uptake of radiolabeled iron together with iodine-labeled transferrin (reviewed in [24]). This allowed for accurate measures of uptake of both iron and transferrin by the brain and, importantly, showed that the transport of radioactive iron through the BBB by far exceeded that of transferrin even a few hours after injection into the peripheral blood. Similar observations were made independently by another research group [25]. This led to the conclusion that iron-containing transferrin is taken up by receptor-mediated endocytosis at the luminal membrane of brain capillaries. In the brain, iron dissociates from transferrin within the slightly acidic environment in the endosomal compartment [26], and iron is transported across the abluminal lipid bilayer of the BECs to the brain, whereas iron-free transferrin is retro-endocytosed back to the luminal membrane [9,24,27].

Counteracting the notion of receptor-mediated endocytosis and retro-endocytosis of transferrin at the BBB, other studies suggested that the iron-containing transferrin may be transferred across the BECs [28,29]. A caveat, in the relevance of these data for understanding iron and transferrin transport at the BBB, shows only transferrin was detected in the brain, hence leaving out the possibility of interpreting the simultaneous transport of iron. However, supporting that transendothelial transport of iron-containing transferrin may occur, observations made on iron and transferrin uptake combined in other studies do not exclude that a minority of iron–transferrin may pass through the BBB, hence simulating transcytosis at the brain endothelium [24].

The expression of transferrin receptor by BECs varies throughout development, with transferrin transport into the brain being higher in the developing brain than at later ages [30,31,32]. Interestingly, although magnitudes lower than that of iron, the transport of transferrin across the BBB is significantly higher than that of albumin [28,29,30], which may be due to higher transcellular trafficking of transferrin. Many more vesicles, typically sized about 70 nm in diameter, are present in BECs of the developing brain [33], so even if only a limited fraction of these vesicles fuse with the abluminal membrane, there would a priori be more vesicles emptying their content into the brain during ontogenesis than in adulthood. Endocytic clathrin-coated vesicles are formed as part of transferrin receptor docking at the luminal side of the BECs. The resulting vesicle forming due to the transferrin attachment will likely also capture fluids from the extracellular space of the luminal side in a non-specific manner, which may explain why albumin also gets transferred through the BBB to a higher degree in the developing brain (Figure 1).

Returning to the attempts to enable transport through the choroid plexus, it should not be overlooked that there is strong morphological and physiological evidence for vesicular transport by transcytosis through this epithelium. Ultrastructurally, tracer studies using peroxidases demonstrate that the choroid plexus epithelium can take up large molecules like horseradish peroxidase (HRP) with transfer from the basolateral to the luminal side eventually leading to release into the ventricles [15,34,35]. The choroid plexus, contrary to the brain endothelium, also contains vesicular structures with albumin, IgG, and transferrin. This is corroborated by the concentration of these proteins being many times higher in the ventricular CSF compared to the interstitial fluid of the brain in the CSF (c.f. [15]). The transport of large plasma proteins could theoretically qualify the blood–CSF barrier as a feasible route for nanoparticles to enter the brain. Counteracting this consideration, CSF of the ventricular system purely distributes substances to the ventricular system and subarachnoid space and excludes the possibility of targeted transport of antibodies and nanoparticles into the brain via transport across the choroid plexus [2].

## 4. Specific Proteins Expressed by Brain Endothelial Cells Enhance the Binding and Uptake of Antibodies from the Circulation

The functional capacity of the brain endothelium to bind and internalize antibodies targeted to the transferrin receptor [18] spawned the idea of using transferrin receptor antibodies to target the brain endothelium, as this would allow conjugated therapeutic molecules to enter the brain [19,21,36,37]. The rationale for injecting transferrin receptor-targeting antibodies is that exogenous transferrin needs to compete with endogenous transferrin of blood plasma, which significantly reduces the likelihood for binding transferrin receptors [36,37]. This is not the case when using transferrin receptor targeting antibodies, which bind to epitopes at the transferrin receptor without interfering with endogenous transferrin. This is advantageous from the physiological point of view, as the brain delivery of iron is not hampered by antibody-targeting of the transferrin receptor [36,37].

The injection of antibodies targeted to the transferrin receptor dramatically increases the brain uptake as compared to non-targeted antibodies [11,36,37]. Noteworthy, injecting antibodies targeting the insulin receptor, also expressed by BECs, similarly allowed for higher uptake in BECs [11,13,19,38]. However, the internalization did not guarantee successful passage across the endothelium. It was later shown that although transferrin receptor-targeted antibodies were internalized in the BECs by receptor-targeting, the antibodies fell short in their capability to pass to the brain parenchyma [16,39]. This observation was explained by antibodies forming covalent binding to the transferrin receptors sufficient to prevent the antibodies from detaching from the receptor. Later, biotechnological advances created the basis for synthesizing mono-specific antibodies lowered with low affinity, and bi-specific antibodies with low avidity. Such antibodies can be constructed by replacement of a single Fab fragment of a monospecific, high-affine, divalent antibody with a Fab fragment able to bind a different molecule. The properties of the modified antibodies counteracted the permeability restraints of the BBB and enabled both uptake and higher transport of transferrin receptor-targeted antibodies into the brain, as verified from their engagement with neurons and proteins deposited extracellularly in the brain [40,41]. Subsequently, these approaches inspired the generation of a plethora of differently designed transferrin receptor-targeting antibodies, all able to transport conjugated biologics across the BBB [41,42,43,44,45,46,47]. Together, they have provided new optimism on how to achieve delivery of therapeutics to the brain, with transferrin receptor-targeted antibodies being now amendable for clinical use in conjugation with enzymes needed for treatment of lysosomal deficiency or being tested in clinical trials against amyloid deposition in Alzheimer’s disease (ClinicalTrials.gov Identifier: NCT05371613; NCT04639050; NCT04573023).

While the development of antibodies entering the brain was generated using antibodies targeting the transferrin receptor and the insulin receptor, it should not be overlooked that targeting other proteins of the brain endothelium has been pursued. The large amino acid transporter (CD98hc), glucose transporter 1 (GLUT1), and basigin (CD147) are particularly rich in their selective expression by the BECs compared to capillaries in the periphery, e.g., lung and liver, and are alternatives for targeting the brain endothelium [19,38,48,49].

## 5. Specific Proteins Expressed by the Brain Endothelium also Facilitate the Binding and Uptake of Targeted Nanoparticles

The plethora of in vivo studies on nanoparticle transport typically omit to characterize the pharmacokinetics that leads to transport through the BBB. They rather focus on pharmacodynamics or therapeutic effects after the particles have undergone transport into the brain., Often the evidence for the latter is scarce and is extrapolated from pharmacological studies, where improvement in behavioral tasks of experimental animals treated with nanoparticles is used as evidence for BBB transport. Many studies determine fluorescent nanoparticles using whole-brain imaging, which prevents the distinction of nanoparticles in BECs versus neurons or glia. Other studies determine changes in protein or gene expression by neurons and glia in dissected brain preparations without taking the expression levels in BECs into consideration, e.g., by analyses of brain capillaries isolated and separated from the remaining brain tissue [50]. 

In spite of the limited number of studies dealing with the uptake and transport kinetics by BEC, a common observation is that the uptake of nanoparticles, e.g., liposomes, gold nanoparticles, or quantum dots, from the circulation is significantly enhanced when conjugated to the transferrin receptor targeted antibodies [51,52,53,54,55]. Comparing the uptake of stealth liposomes in the mouse brain with or without conjugation to proteins putatively targeting BECs, only antibodies targeting the transferrin receptor (clone RI7217) enhanced the liposomal binding and uptake by the brain endothelium [52]. The uptake of RI7217-conjugated liposomes was almost two-fold higher compared to binding with endogenous transferrin or un-conjugated liposomes in brain capillaries isolated from the brain 12 h post injection [52]. Independent studies in the rat [51,53] and mouse [54,55] also concluded that targeting the transferrin receptor using high-affine anti-transferrin receptor antibodies leads to preferential accumulation of liposomes within BECs (Figure 2). Given this evidence, it stands out as somewhat puzzling that reports continuously occur addressing targeting attempts to the transferrin receptor at BECs using only transferrin and not the antibody.

The uptake of RI7217-conjugated liposomes at the BBB is significantly higher compared to stealth liposomes conjugated with non-immune IgG [52]. In contrast, liposomes conjugated to proteins putatively capable of targeting BECs, e.g., (i) cross-reacting material (CRM) with affinity for an endogenous diphtheria toxin receptor; (ii) angiopep-2 with affinity for LRP-1; (iii) COG133 with affinity for apolipoprotein E, all failed to exhibit higher uptake compared to non-immune IgG-conjugated liposomes [52]. In particular, the observations made on liposomes targeting the diphtheria toxin receptor, LRP-1, or apolipoprotein E were discouraging [52], although earlier studies indicated that these targets were relevant for nanoparticle uptake in BECs [56,57,58]. Studies using unconjugated antibodies targeting LRP-1 also failed to prove that LRP-1 was a viable target for specific uptake by the brain endothelium [19]. It is feasible that the widespread expression of the aforementioned targets in peripheral vasculature may reduce the extent of liposome availability for uptake at BECs.

The number of antibodies present on the surface of antibody-functionalized gold nanoparticles and cargo-loaded stealth liposomes influenced the targeting to BECs in vivo. Hence, the highest density out of a selection of different densities (0.15, 0.3, and 0.6 ∗ 10^3^ antibodies/μm^2^) led to the highest binding and uptake [59]. Using gold nanoparticles conjugated with targeting antibodies with different affinity for the transferrin receptor [40], or lowering the avidity inversely led to higher uptake of targeted nanoparticles both in vivo in the adult mouse and in vitro in isolated primary mouse BECs from adult mice [54] (Figure 3). Examining the influence of the avidity of the targeting antibodies, using bispecific antibodies targeting both the transferrin receptor and amyloid beta (i.e., mono-valent binding to the transferrin receptor, this approach resulted in higher binding and uptake when compared to low-affine, bivalent monospecific antibodies both in vivo and in vitro [54].

The uptake of nanoparticles may be further enhanced by changing the shape of nanoparticles, provided they are constructed by a relevant material [60,61]. In vitro studies showed that rod-shaped polymeric nanoparticles targeted to the transferrin receptor underwent seven-fold higher uptake compared to spherical particles [60], clearly warranting further pursuit in vivo. The nanoparticles’ zeta potential is also important with slightly negative potential being optimal for uptake [11,12,13]. When present in blood plasma, nanoparticles tend to absorb blood-circulating proteins forming a so-called protein corona so significant that it may sterically block for binding of the targeting protein to its receptor [61]. Previous research on the liposomal protein corona may have suffered from significant methodological limitations making this issue too problematic as justified in more controlled experiments, showing that contaminating proteins may have interfered with the analysis of the protein corona [62,63,64]. To prevent a potential unwanted influence of protein corona on the targeting potential of the nanoparticles, the nanoparticles can be conjugated to their targeting antibody bridged in-between by PEG molecules, which simultaneously will limit their likelihood of being taken up in the reticuloendothelial system (RES) [9,10]. 

The possibility of using the transferrin receptor for targeting nanoparticles to the BECs has also been pursued in studies using dual targeting approaches in which transferrin is linked to the surface of nanoparticles in conjunction with other peptides. Studies were mainly performed in vitro, with additional biodistribution studies in vivo using nanoparticles conjugated with transferrin and cell-penetrating peptides (CPPs), or rabies virus glycoprotein (RVG) [65,66,67]. Although these approaches bear great potential, they warrant direct comparisons with antibodies targeting the transferrin receptor for efficient drug delivery to the brain. Other strategies examined the transferrin receptor targeting peptide T7 (aka T7-LS) bound to the surface of liposomes containing the chemotherapeutic vincristine and reported a significantly higher pharmacological effect compared to the targeting of liposomes bound to other transferrin receptor targeting peptides B6 and T12 [68].

Despite being available for experimental use for more than three decades, the transferrin receptor is by far still considered the most relevant target for nanoparticle drug delivery. It is only a few other candidates, such as CD98, GLUT1, basigin (CD147), and the insulin receptor that were being taken into consideration as an alternative, but other than the insulin receptor, these receptors remain unexploited for their targetability to BECs with regard to nanoparticles [19]. 

## 6. Anti-Transferrin Receptor Antibodies Weakened in Affinity or Lowered in Avidity Facilitate Nanoparticle Transport through BECs

Anti-transferrin receptor-targeted monoclonal antibodies weakened in affinity for the receptor readily undergo transport across the BBB in a dose-dependent manner, becoming detectable in neurons [40]. Bispecific antibodies, with one domain targeting the transferrin receptor at high affinity and the other domain directed towards a putative therapeutic target relevant for treating Alzheimer’s disease, i.e., beta-secretase 1, lead to reduction of amyloidogenic peptide formation in the brain [41]. Unfortunately, studies have not yet been performed to examine the efficacy of targeted nanoparticles with respect to repeated or chronic dosing regimens. Such evaluations would help to understand the amounts of therapeutics encapsulated within nanoparticles that can be accumulated inside the brain.

Antibodies with low affinity for the transferrin receptor seem to follow a cellular route identical to that of high-affine antibodies, as prior exposure to high-affine antibodies leads to reduced transport of low-affine antibodies across the BBB both in vivo and in vitro [69]. The two different antibodies differ in that antibodies with low affinity are not directed towards lysosomes to the same extent as antibodies with high affinity [69]. Further information on intracellular transport relying on anti-transferrin receptor antibodies comes from the study on the subcellular distribution of anti-transferrin receptor antibody-conjugated gold nanoparticles [54] (Figure 4). This revealed gold particles in BECs, and when conjugated with low-affine or low-avidity antibodies, the gold particles were also detected in neurons, further arguing for transport across BECs. The targeted gold nanoparticles were apparent in BECs in clearly identifiable vesicular structures, which might represent sorting endosomes and lysosomes. The study did not identify gold nanoparticles fusing with the abluminal membrane, but this does not exclude the transcellular transport of the nanoparticles through the BBB. This uncertainty comes from the observation of a low number of sequestered particles, and that the electron microscopy data were collected from thin sections, typically less than 100 nm. In turn, isolated mouse BECs arranged in hanging cell culture inserts with defined BBB properties revealed transcellular transport of gold nanoparticles when conjugated with low-affine or low-avidity antibodies, which supports the observation of nanoparticle transport across the BBB. Notably, the electron microscopy studies did not show signs of obstructive accumulation of the gold nanoparticles sized approximately 75 nm near the basement membrane, which may be a major restraint for nanoparticle trafficking in the brain’s extracellular volume after release at the abluminal side of the BECs [9].

## 7. A Mechanistic Approach to an Understanding of Trafficking of Transferrin Receptor-Targeting Liposomes Based on Studies of Iron-Transferrin and Unconjugated Anti-Transferrin Receptor Antibody Trafficking in BECs

The paucity of studies examining intracellular trafficking of nanoparticles in BECs in vivo limits the available information about their transport mechanisms. Some lessons may be learned from comparison of endogenous transferrin and unconjugated, targeted anti-transferrin receptor antibodies. Although this difference obviously should be taken into account, the low-affinity antibodies and low-affinity antibodies conjugated to PEGylated liposomes share great similarities in transport through the BBB with unconjugated low-affinity and low-affinity anti-transferrin receptor antibodies [40,54].

### 7.1. Blood to Endothelium Transport

Endogenous transferrin enters the BECs after the interaction with the transferrin receptor on the luminal surface, facilitating subsequent formation of clathrin-coated pits, and eventually, formation of endosomes [9,16,24]. The endosomes have a slightly acidic pH, which promotes detaching iron from transferrin [9,16]. Consequently, the unbound iron can cross the endosomal membrane via divalent metal transporter 1 (DMT1), which makes the iron available in the cytosol [9,25]. In parallel, the iron-depleted apo-transferrin residing inside the endosome loses its affinity for the transferrin receptor, and is thought to undergo retro-endocytosis to the luminal surface of the endothelial cell [9,16].

### 7.2. Endothelium to Brain Transport

Whereas the docking and endosomal formation relate to the affinity of the transferrin receptor, it can be argued that the intracellular trafficking of the endosomes follow routes that occur independently of the luminal receptor internalization [9,16,51]. BECs contain RAB4 and RAB7 proteins specific for early and later sorting endosomes [70,71]. The abluminal membrane of the BECs also contains the protein TSG101 (tumor susceptibility gene 101), which takes part in exocytosis. This suggests that BECs contain organelles fully capable of handling transferrin receptor-containing vesicles that present themselves initially as early forming endosomes with the capacity to fuse with late endosomes, eventually leading to fusion with the abluminal membrane and exocytosis [72]. Therefore, the process of sorting transferrin receptor-containing vesicles is likely to be in two ways: Both morphological and pharmacological studies favor receptor-mediated endocytosis taking place at the luminal side leading to formation of early endosomes. This is followed by a vaguely understood trafficking of late endosomes directed towards fusion with the abluminal surface [9,16]. 

A morphological approach to detect transferrin at the ultrastructural level in rats subjected to the brain in situ perfusion failed to detect HRP-conjugated iron-transferrin near the abluminal side, which, in turn, was outnumbered by the presence of HRP-transferrin in multiple vesicular-like structures near the luminal side of the BECs [58]. However, as mentioned earlier, in the developing brain BECs are enriched in vesicles involved in transcellular trafficking. This, together with the observation that the developing brains have relatively higher expression of transferrin receptors [30,31], could account for directed transcellular trafficking of iron-transferrin-containing vesicles through the BECs. Interestingly, intracarotid perfusion with OX26-conjugated colloidal gold enabled detection at the abluminal side of the brain endothelium [58]. Although this study did not quantify the transport of the gold-labeled OX26, the appearance near the abluminal side may represent transferrin receptor-containing vesicles available for fusion at the abluminal side. A conclusion may be that as only a minor fraction of iron-transferrin within the transferrin-containing vesicles moves towards the abluminal side of the BEC, only a minor fraction of such vesicles including their content are released at the abluminal side of the BECs.

The binding to high-affine anti-transferrin receptor antibodies also leads to the formation of endocytotic vesicles that mainly localize near the luminal membrane [9,16]. The uptake and transport of high-affine, anti-transferrin receptor antibodies within the BECs are likely to follow the same route as that of iron-transferrin. However, differences may occur, as the binding of the high-affine antibody to the transferrin receptor is thought not to be reversible, leading to accumulation of the antibody within the endosomal-lysosomal compartment [16]. Noteworthy, later studies addressing the fate of the anti-transferrin receptor antibodies showed that the complexes are incorporated in lysosomes rather than being directed towards release at the abluminal membrane [69]. A study in a mouse model with human transferrin receptors revealed that high-affine anti-transferrin receptor antibodies could undergo transport across the BBB [43]. However, it is very difficult to compare the consequences of binding affinity of anti-transferrin receptor antibodies between species concerning the capability to undergo transport at the BBB. The mechanisms that enable high-affine and low-affine antibodies to detach from the transferrin receptor within the endosome is poorly understood, but possibly, the acidic environment in the endosome facilitates the detachment of antibodies from the receptor. Even if a minor fraction of antibodies bound to the transferrin receptor is released in the acidic endosomal environment, a fraction of the high-affinity antibody would be released and would move further into the brain’s extracellular space.

It goes beyond doubt that the optimal transport of targeted nanoparticles at the BBB depends on the binding of anti-transferrin receptor antibodies to the endothelial surface [68]. The transport of nanoparticles through BECs appears to depend on the affinity or low avidity of the anti-transferrin receptor antibody. Nanoparticles, therefore, may undergo transport through BECs similar to therapeutically active molecules like enzymes or other proteins directly conjugated to anti-transferrin receptor antibodies [73] (Figure 5).

## 8. Post-Capillary Venules Denote an Alternate Route for Transport

The use of 2-photon microscopy (2PM) for in vivo studies has revolutionized the concept of how the brain works and has recently entered the field of the BBB research [74]. 2 PM visualizes fluorescently labeled molecules in the brain with a sub-micron spatial resolution and allows for visualization of individual nanoparticles in blood vessels to a depth of 600 µm below the pial surface in vivo [74]. Recent studies using 2PM on the transferrin receptor-targeted liposomes demonstrated how endothelial cells of both brain capillaries and post-capillary venules in contrast to arterioles handle transferrin receptor-targeted liposomes at the level of a single nanoparticle [14] (Figure 6). The major finding was that the liposomes targeted with high-affine RI7217 were released to the brain almost exclusively from endothelial cells of the post-capillary venules, with negligible results.

One study showed the contribution of endothelium at capillaries [14], which is scientifically provocative, considering previous investigations using this antibody to study transport across the brain endothelium. The release of the liposomes from the post-capillary venules is surprising, and the mechanisms and explanation for this observation will need further study. The barrier formed by the endothelial cells of the post-capillary venules is less tight at the venous side with respect to the number of tight junctions compared to BECs [75]. This could principally allow nanoparticles to enter the brain via paracellular transport, but counteracting this notion, the targeted nanoparticles were clearly taken up by the endothelial cells of the post-capillary venules before entering the brain on the abluminal side, verifying transendothelial transport of the targeted liposomes. In all, the density of transferrin receptor-targeted nanoparticles, being highest in capillaries, did not translate to efficient transport of nanoparticles to the brain. Thus, the BBB is highly heterogeneous regarding transport mechanisms, and, in particular, the ability to transcytose large constructs across the BBB [14].

The study of Kucharz et al. [14] clearly opens for novel considerations on the transport of targeted nanoparticles to the brain. An interesting consideration related to prior studies demonstrating targeted liposomes within the brain [11,12,13,54] is that they could also have passed into the brain via this hitherto overseen route at post-capillary venules. 

## 9. Targeting Nanoparticles to the Brain Endothelium in Pathological Conditions

The use of targeting approaches to promote enhanced drug delivery to the brain in pathology is only coming of age. Targeting approaches in conditions with cerebral pathology can roughly be separated into attempts to treat the brain in acute and chronic conditions. In acute conditions, e.g., ischemic stroke and traumatic brain injury, liposomes are advantageous among nanoparticles because they can be formulated to contain degradable lipids with enzymes like matrix metalloproteinases and phospholipases known to be increased in expression and released from the brain in acute pathology [76,77,78,79]. In terms of delivery to the brain, the liposomes will easily enter the affected brain regions as the vasculature is deteriorating, leading to the opening of the BBB. The liposomes are expected to start to degrade once they enter areas of the brain with a raised expression of, e.g., matrix metalloproteinases. This approach does not demand the liposomes to be functionalized by conjugation to a targeting antibody, but the targeting approach could allow for more widespread uptake of the liposomes in the surrounding areas of a central pathology, e.g., in ischemic stroke, the penumbra zone, where the BBB often remains intact [80]. Targeting to the BECs could enhance the liposomal delivery, which might allow, e.g., enhanced pharmacological preconditioning using focused ultrasound and microbubble treatment [81]. However, it should be noted that many acute conditions are also associated with obstructed blood flow due to pericyte-mediated constriction of capillaries, as observed, e.g., post stroke [82,83]. As such, the obstructed blood flow may limit the ability of the blood circulation to deliver the liposomes to the relevant, damaged areas in the brain. 

Recent efforts aim to utilize a dual targeting approach using antibodies targeting the transferrin receptor and intercellular adhesion molecule-1 (ICAM1), both conjugated to liposomes. This resulted in higher binding compared to liposomes targeted to only transferrin receptor or ICAM1 antibodies alone [84]. This approach led to enhanced delivery of liposomes encapsulating the anti-inflammatory cytokine TNF-alpha in a model of acute brain inflammation [84].

In chronic brain disorders, like Alzheimer’s disease and Parkinson’s disease, the brain does not increase the expression of liposome-degradable enzymes like metalloproteinases and phospholipases to the same extent as seen in acute neuropathology, and therefore, the strategy of substrate degradable liposomes may not apply. Instead, it may be advantageous that the BEC expression of the transferrin receptor remains unchanged in Alzheimer’s disease [85], which further justifies attempts to target transferrin receptors expressed by BECs in the neurodegenerative brain expected to have a near intact BBB.

## 10. Conclusions

BECs are practically incapable of uptake of native liposomes from the blood, which requires the addition of targeted molecules. Experience gained from different targeting approaches justifies the choice of antibodies targeting the transferrin receptor for targeted delivery of nanoparticles to BECs. Liposomes functionalized by conjugated anti-transferrin receptor antibodies are taken up by endothelial cells at both brain capillaries and post-capillary venules. Modulating the number of transferrin receptor-targeted antibodies shows that limiting the number of targeting antibodies conjugated to liposomes reduces uptake in BECs. In comparison, lowering the affinity of the targeting antibodies or implementing bispecific antibodies with low avidity increases transport by BECs into the brain parenchyma. With increasing evidence of successful preclinical trials and advances in biochemical and analytical approaches, the transferrin receptor-targeted nanoparticles have great promise for future use in drug delivery as they evidently pass the BBB.

## Figures and Tables

**Figure 1 pharmaceutics-14-02237-f001:**
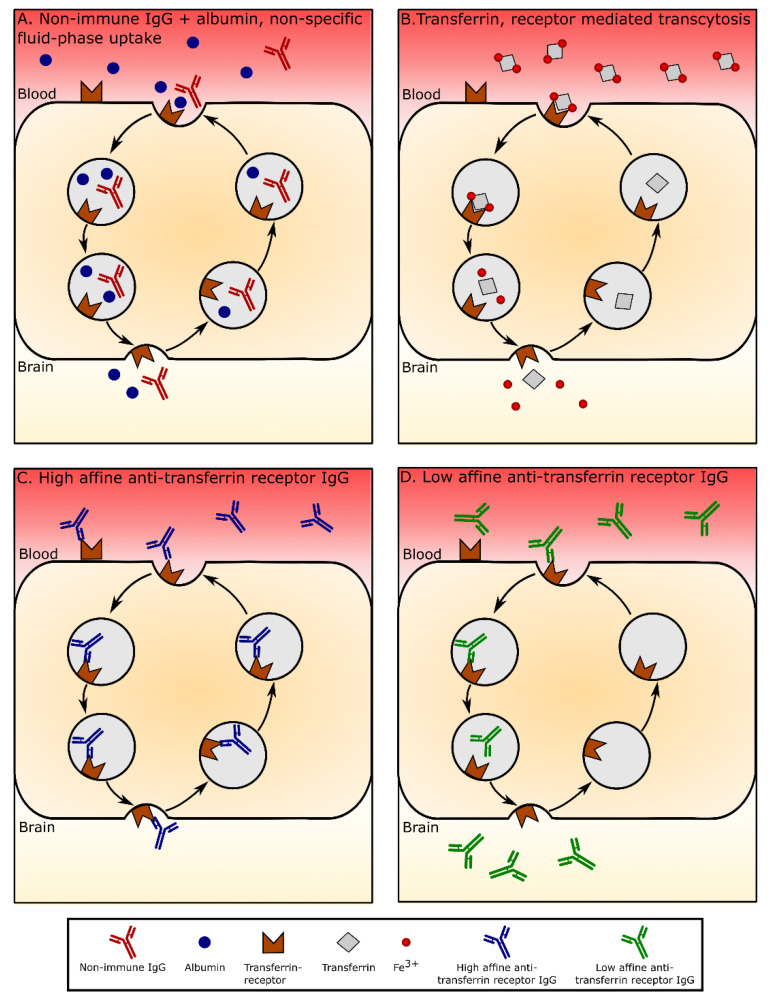
Illustration of transport of the three major plasma proteins and targeted anti-transferrin receptor antibodies within brain capillary endothelial cells (BECs). (**A**) Albumin and non-immune IgG enter the BEC non-specifically by fluid-phase uptake. They may undergo release from the BEC at the abluminal side, although the quantitative evidence is scarce and clearly shows that this mean for entry of albumin and non-immune IgG to the brain is negligible. (**B**) Similarly, the transcytotic transport of iron-containing (holo-transferrin) is also negligible and opposed by the release of iron from transferrin due to the lower pH of the endocytosis vesicle. Transport through the BECs may occur to a higher extent in the developing brain where transferrin receptor expression by BECs is far-fold higher than in the adult brain (see body text). (**C**) Transport of high-affine IgG targeted to the transferrin receptor. Transport of this antibody through BECs is negligible. (**D**) Transport of low-affine IgG targeted to the transferrin receptor. Transport of this antibody through BECs is enabled by the detachment of the antibody from the transferrin receptor in late endosomes, which enables this antibody to undergo transport into the brain. Modified from [15].

**Figure 2 pharmaceutics-14-02237-f002:**
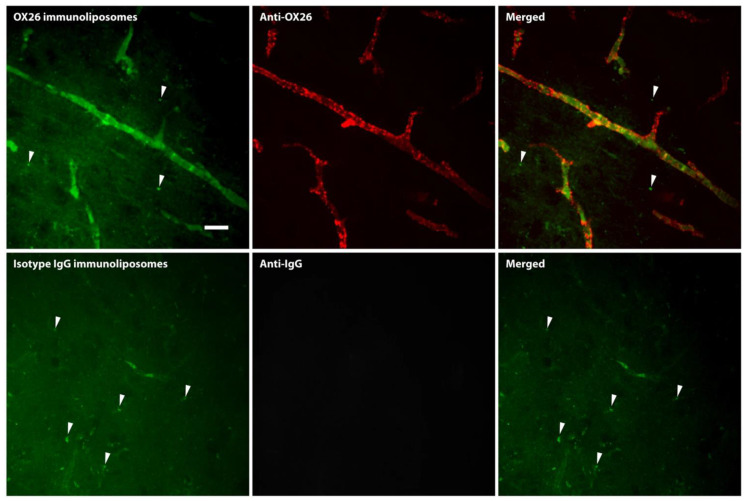
Uptake of fluorescently labeled immunoliposomes conjugated with high-affine IgG targeted to the transferrin receptor (OX26) in brain capillaries in vivo in the rat as revealed using spinning disk confocal microscopy. The OX26 immunoliposomes associate to brain microvessels. Immunohistochemical detection of the OX26 of the immunoliposome similarly reveals that the immunoliposome and its binding antibody accumulate in the brain capillaries. Scale bar = 20 µm. Modified from [53].

**Figure 3 pharmaceutics-14-02237-f003:**
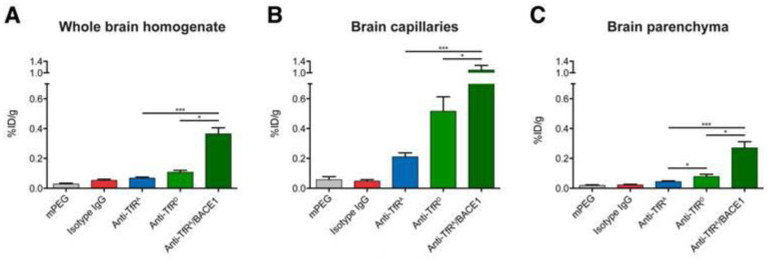
Uptake in the adult mouse brain of gold-labeled nanoparticles (AuNPs) targeted to the BBB by anti-transferrin receptor antibodies varying in affinity. Identical antibodies were studied prior for BBB transport without conjugation [44]. (**A**) In whole brain homogenates, there is a clear distinction between the different transferrin receptor (TfR)-targeted variants with respect to their accumulation. (**B**) The TfR-targeted AuNPs accumulate in the capillary fraction with 0.2, 0.5, and 1.1%ID/g for anti-TfR^A^, anti-TfR^D^, and anti-TfR^A^/BACE1. (**C**) In fractions containing brain parenchyma, detection of AuNPs indicates transport across the BBB. Accumulation is mainly seen for the low-affinity anti-TfR^D^ compared to the high-affinity anti-TfR^A^ variant AuNPs. Anti-TfR^A^/BACE1 AuNPs are superior to the other TfR-targeted variant AuNPs with a mean parenchymal accumulation of 0.23% ID/g compared to 0.04 and 0.08% ID/g for anti-TfR^A^ and anti-TfR^D^, respectively. Modified from [54]. Data are presented as mean ± SEM (n = 7–8, Kruskal-Wallis test with Dunn’s multiple comparisons post hoc test) with * *p* < 0.05 and *** *p* < 0.001. %ID/g: percentage of injected dose per gram.

**Figure 4 pharmaceutics-14-02237-f004:**
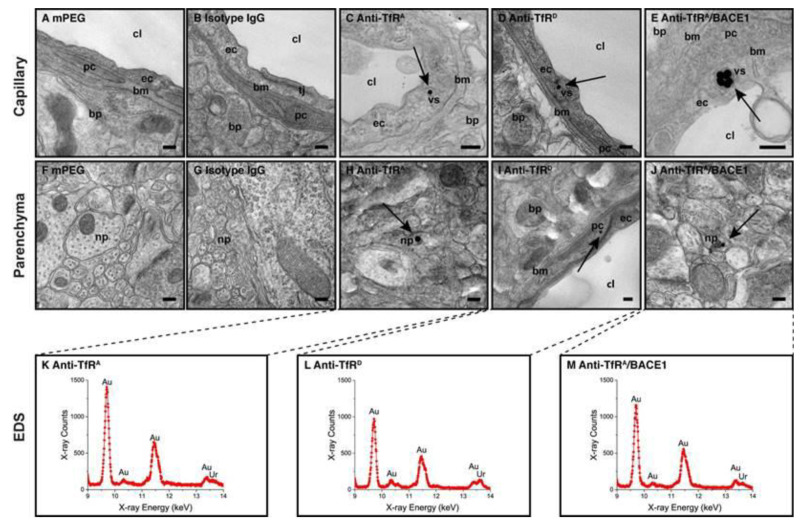
Detection of anti-transferrin receptor IgG conjugated gold-labeled nanoparticles (AuNPs) using transmission electron microscopy (TEM) in a normal adult mouse brain. Anti-transferrin receptor antibodies vary in affinity. (**A**,**B**) AuNPs are not detected in brain capillaries of mice in the mPEG (no IgG added) or isotype (non-immune) IgG groups. (**C**–**E**) In contrast, the AuNPs targeted with anti-transferrin receptor IgG are found in BECs (arrows). The AuNPs are detected in BECs confined to vesicular structures, suggesting receptor-mediated endocytosis as the uptake mechanism. (**F**,**G**) In brain parenchyma, AuNPs are not detected in mice in the mPEG or isotype (non-immune) IgG groups. (**H**–**J**) AuNPs are seen in brain parenchyma of mice treated with all transferrin receptor (TfR)-targeted variants, among which they are most easily detected in the anti-TfRA/BACE1 group (**J**). The sites for transport of the AuNPs may derive from transport across either BECs or post-capillary venules (see text body). All AuNPs detected in the brain parenchyma were analyzed using energy-dispersive X-ray spectroscopy (EDS) to validate the true presence of gold in the electron-dense points (**K**–**M**). Scale bars depict 200 nm. bp: brain parenchyma; bm: basement membrane; cl: capillary lumen; ec: endothelial cell; np: neural process; pc: pericyte; tj: tight junction; vs: vesicular structure. Modified from [54].

**Figure 5 pharmaceutics-14-02237-f005:**
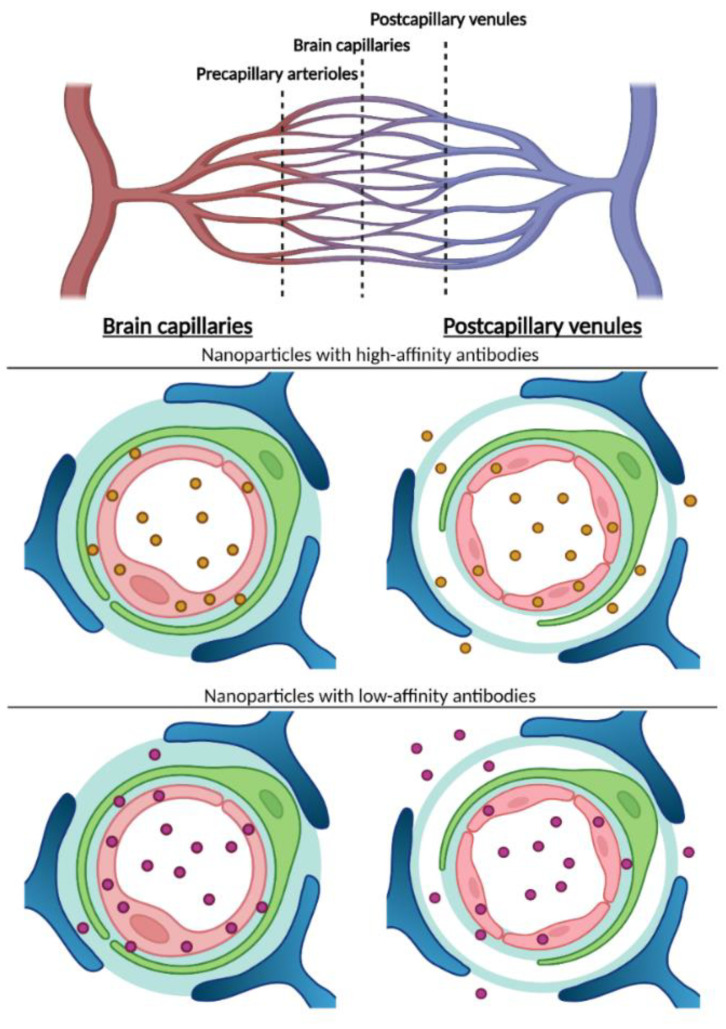
Working model of transcytosis-mediated nanoparticle delivery to the brain. Recent study provides evidence that high-affine anti-transferrin receptor IgG conjugated to liposomes mainly undergo transport into the brain at the site of post-capillary venules [14]. This observation counteracts that of unconjugated antibodies that passes the BBB at the site of brain capillaries when designed to be low in affinity or avidity [42]. Targeting the latter antibodies to gold nanoparticles leads to capillary transport in vitro using isolated brain capillaries [58] suggesting that accumulation of targeted nanoparticles in brain parenchyma in vivo may occur via transport across capillaries as well. Studies concerning transport into brain across post-capillary venules using transferrin receptor-targeted low-affine antibodies (lower right) have not been performed but can be stipulated to lead to enhanced transport compared to the use of corresponding antibodies with high affinity. Red bullets: high-affine anti-transferrin receptor IgG conjugated liposomes. Yellow bullets: low-affine or low-avidity anti-transferrin receptor IgG conjugated to liposomes. Drawing created with BioRender, inspired and modified from [14].

**Figure 6 pharmaceutics-14-02237-f006:**
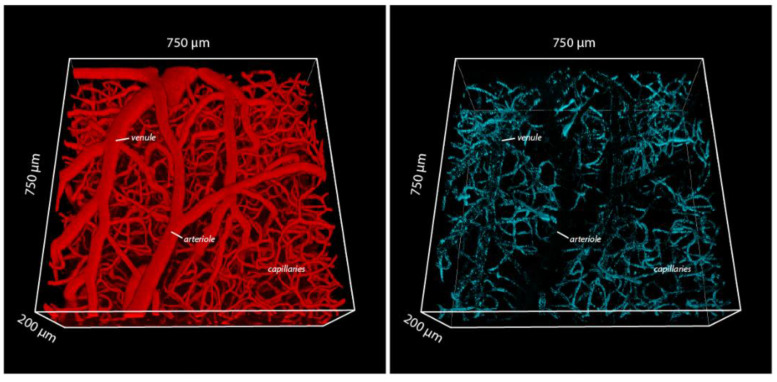
Example of in vivo 2PM data, showing three-dimensional reconstruction of cortical microvasculature (**left**), and the distribution of intravenously-injected transferrin receptor-targeted liposomes residing at the BBB interface (**right**). The transferrin receptor-targeted liposomes associate primarily to capillaries, then venules, but are absent in arterioles. The images were collected 2 h post injection. Modified from [14].

## Data Availability

None.

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
