# Peer review of "Blood–Brain Barrier Transport of Transferrin Receptor-Targeted Nanoparticles"

_pharmaceutics, 2022, doi:10.3390/pharmaceutics14102237_

Round 1
Reviewer 1 Report
The document is very interesting, actual and well written, however it is not clear what type of document it is (review or book chapter). If it is a review, the title should mention it . In this case it may be a narrative review, so the search engines (data bases) must be named, as well as the time period (years) and the words/sentences used to do the search. That information should complement what was written from line 72 to 77.
Some minor suggestions:
26: “ CSF, cerebrospinal fluid barrier” correct to cerebrospinal fluid (as in line 43)
94: correct “superseded “ to increased ??
157: correct “being many folds higher” to “being many times higher”
176: Correct/rephrase “However, the internalization did guarantee successful passage across the endothelium.”
179: Correct/rephrase “This observation was explained by antibodies forming covalent binding to the transferrin receptors sufficiently strong to prevent the antibodies from detaching from the receptor” Are there week covalent binding?????
204: Correct “Often studies determine” to “several/many….”
246: “as earlier studies” or “although earlier studies”???
300: Correct “Although been” to “Despite being”
462: remove “pathology” or consider pathological conditions? Or unhealthy conditions?
472: correct/rephrase “in the areas surrounding areas with central pathology” to “in the surrounding areas of a central pathology”???
491: correct “necessitates” to “requires”
Author Response
Reviewer I
Replies to Reviewer I
- It is now indicated in the top header that the manuscript is a review paper. We have based the review on the most pertinent papers published in the literature using a broad search.
- We have included virtually all the minor suggestions made by Reviewer I and appreciate these suggestions.
Author Response
Reviewer II
Replies to Reviewer I
- We politely refrain from including more references on LRP-1 for BBB targeting: We have already included three references, and most importantly we have included two studies that both failed to prove evidences of LRP-1 targeting for BBB transport being it antibodies directed to LRP-1 or nanoparticles conjugated with LRP-1 ligands. Therefore, we find it justified what we have already written in the manuscript. The suggested references made by reviewer I do not include evidences of BBB transport at the morphological level to clearly determine the presence of particles inside the brain, and they did not include brain capillary purification experiments to separate the capillaries form the remaining brain tissue.
- The reviewer comments on the use of transferrin receptor-targeting peptides: We have included what we find the most relevant for BBB transport, and this is in fact the only evidence published so far, we are including. So, we are in fact already including the relevant literature.
- We have revised the manuscript for mistaken punctuations.